

# Electric manipulation of domain walls
# in magnetic Weyl semimetals via the axial anomaly

**Julia D. Hannukainen[1], Alberto Cortijo[2,3], Jens H. Bardarson[1] and Yago Ferreiros[1,4]**

**1** Department of Physics, KTH Royal Institute of Technology, Stockholm, 106 91, Sweden
**2** Departamento de Física de la Materia Condensada,
Universidad Autónoma de Madrid, Madrid E-28049, Spain
**3** Condensed Matter Physics Center (IFIMAC), Madrid E-28049, Spain
**4** IMDEA Nanociencia, Faraday 9, 28049 Madrid, Spain

## Abstract

We show how the axial (chiral) anomaly induces a spin torque on the magnetization in magnetic Weyl semimetals. The anomaly produces an imbalance in left- and right-handed chirality carriers when non-orthogonal electric and magnetic fields are applied. Such imbalance generates a spin density which exerts a torque on the magnetization, the strength of which can be controlled by the intensity of the applied electric field. We show how this results in an electric control of the chirality of domain walls, as well as in an improvement of the domain wall dynamics, by delaying the onset of the Walker breakdown. The measurement of the electric field mediated changes in the domain wall chirality would constitute a direct proof of the axial anomaly. Additionally, we show how quantum fluctuations of electronic Fermi arc states bound to the domain wall naturally induce an effective magnetic anisotropy, allowing for high domain wall velocities even if the intrinsic anisotropy of the magnetic Weyl semimetal is small.

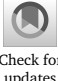

# 1 Introduction

Manipulating magnetization with electrons is at the heart of spintronics, as this allows for the development of magnetic technologies for information storage, computation, and sensing [1,2]. Since the development of the physics of multiferroics, there is a large interest in the electric manipulation of magnetic structures. Different routes include the use of magnetoelectric coupling [3] or spin-polarized currents [4–6]. More recently, new possibilities have appeared with the advent of metallic topological states of matter, which started with studies on electric control of magnetization by the surface states of topological insulators [7–15], and continued with works on spin-to-charge conversion [16] and spin torques [17,18] in magnetic Weyl semimetals. Both systems share a strong spin-momentum locking. As of today, there is an increasing number of experimental studies on magnetic Weyl semimetals [19–22], some of which have reported the presence of domain wall structures [23, 24]. Simple movement of such domain walls can in principle activate exotic Weyl semimetal physics [25], such as the axial anomaly [26–28].

In spin rotation invariant systems, the Zeeman coupling between the magnetization and electron spins is equivalent to an $SU(2)$ gauge field. When interband transitions between majority and minority spin carriers are suppressed, this local $SU(2)$ symmetry can be further projected into a $U(1)$ sector. This projection leads to the appearance of effective electromagnetic fields that give rise to unconventional electric responses, like the topological Hall effect [29] or electric currents induced by the motion of magnetic textures [30].

Quite generically, the Weyl semimetallic phase appears in systems with large spin-orbit coupling, and the aforementioned spin rotation invariance is absent. For those Weyl semimetals that are also magnetic, there are no spin majority and minority states, so any adiabatic projection of the Zeeman coupling over one spin species is not possible. It happens, however, that the magnetization $\boldsymbol{m}$ [1] couples to the spin density operator for the Weyl states, as described by the Hamiltonian

$$H = \Psi^{\dagger} \left( v_F \tau_z \otimes \boldsymbol{\sigma} \cdot \partial + \Delta \mathbb{1} \otimes \boldsymbol{\sigma} \cdot \boldsymbol{m} \right) \Psi, \tag{1}$$

where $\boldsymbol{\tau}$ and $\boldsymbol{\sigma}$ are Pauli matrices acting in chirality-, and spin- space respectively, $\mathbb{1}$ is the identity, $v_F$ is the Fermi velocity and $\Delta$ is an effective coupling. Therefore the magnetization acts as an effective $U(1)$ axial vector field $\boldsymbol{A}_5 = \Delta \boldsymbol{m}/e v_F$ [31], coupling with opposite sign to the two chiralities. In view of this identification, the last term in Eq. (1) can be written as $\boldsymbol{J}_5 \cdot \boldsymbol{A}_5$, and the spin density operator $\boldsymbol{S} = \Psi^{\dagger} \boldsymbol{\sigma} \Psi = \boldsymbol{J}_5/e v_F$ plays the role of an axial current density. It is important to note that the curl of the magnetization now plays the role of an axial magnetic field: $\boldsymbol{B}_5 = \frac{\Delta}{e v_F} \boldsymbol{\nabla} \times \boldsymbol{m}$.

The above identifications are central to the manipulation of the spin dynamics of magnetic Weyl states by electromagnetic fields in a couple of ways: a) through axial current responses under the effect of electromagnetic or axial fields (see, e.g., Fig. [2] of Ref. [32]), and b) through charge accumulation at magnetic domain walls due to the presence of Fermi arc

---

[1] We take $\boldsymbol{m}$ as a unit vector representing the magnetization direction.

states [33]. This last case directly implies that one can manipulate domain walls by applying electric fields, which is particularly relevant in the context of the general interest in electric control manipulation of magnetic structures in multiferroics.

The effect of electrons on the magnetization dynamics can be described by the presence of a spin torque acting on the magnetization [6, 34], whose dynamics are described by the Landau-Lifshitz-Gilbert equation:

$$\frac{d\boldsymbol{m}}{dt} = \gamma \boldsymbol{B} \times \boldsymbol{m} + \alpha \boldsymbol{m} \times \frac{d\boldsymbol{m}}{dt} + \boldsymbol{T}_e, \tag{2}$$

where $\gamma$ is the electron gyromagnetic ratio, and $\alpha$ is the Gilbert damping constant. In the case of Weyl fermions, the spin torque is

$$\boldsymbol{T}_e = \frac{\Delta}{\rho_s} \boldsymbol{m} \times \boldsymbol{S} = \frac{\Delta}{e v_F \rho_s} \boldsymbol{m} \times \boldsymbol{J}_5, \tag{3}$$

where $\rho_s$ is the number of local magnetic elements per unit volume [17]. In Ref. [17] it is theoretically described how an axial current $\boldsymbol{J}_5$ can be induced in a magnetic Weyl semimetal by an external electric field $\boldsymbol{E}$ through a mechanism similar to the conventional Hall effect, mediated by the electronic states around the Fermi surface, but induced by an axial magnetic field $\boldsymbol{B}_5$:

$$\boldsymbol{J}_5 = \sigma_H \boldsymbol{B}_5 \times \boldsymbol{E}. \tag{4}$$

The Hall coefficient $\sigma_H$ now is a function of the effective cyclotron frequency $\omega_c \sim |\boldsymbol{B}_5| \sim |\boldsymbol{\nabla} \times \boldsymbol{m}|$.

In this work, we propose a new mechanism for the generation of an electric-field-induced spin torque in Weyl semimetals. This mechanism is based on the combination of the axial anomaly and the axial separation effect [35–37]. The axial anomaly represents the generation of axial charge through the application of non-orthogonal electric and magnetic fields [26–28]

$$\partial_t n_5 = \frac{e^2}{2\hbar^2 \pi^2} \boldsymbol{E} \cdot \boldsymbol{B}, \tag{5}$$

where the axial number density, $n_5 = n_R - n_L$, is the difference in the number of left- and right-handed fermions. The axial separation effect is the mechanism by which an axial current is induced by an axial magnetic field $\langle \boldsymbol{J}_5 \rangle \sim \mu_5 \boldsymbol{B}_5 = \mu_5 \boldsymbol{\nabla} \times \boldsymbol{m}$, where $\mu_5 = (\mu_R - \mu_L)/2$ is the axial chemical potential representing the difference of chemical potentials for the two chiralities. This is the axial counterpart of the chiral magnetic effect [38], and is derived in section 3. Putting both mechanisms together, the axial anomaly generates a stationary axial chemical potential $\mu_5 \sim \tau \boldsymbol{E} \cdot \boldsymbol{B}$ after introducing chirality flipping scattering events represented by $\tau$ [38]. Then, the axial separation effect gives $\langle \boldsymbol{J}_5 \rangle \sim \tau (\boldsymbol{E} \cdot \boldsymbol{B}) \boldsymbol{\nabla} \times \boldsymbol{m}$. The spin torque generated by the axial current can then be written as $\boldsymbol{T}_e \sim \tau (\boldsymbol{E} \cdot \boldsymbol{B}) \boldsymbol{m} \times \boldsymbol{\nabla} \times \boldsymbol{m} \sim -\tau (\boldsymbol{E} \cdot \boldsymbol{B})(\boldsymbol{m} \cdot \boldsymbol{\nabla}) \boldsymbol{m}$. This results in an electric control of the torque on the magnetization through the axial anomaly. We show how one can electrically manipulate the chirality (the manner in which the magnetization changes between two domains [39]) of domain walls through this mechanism. Achieving controllable ways for chirality manipulation could be important for the development of spin wave based logic gates [40–42]. We also show how such an electrically modulated spin torque can improve the domain wall dynamics by delaying the onset of the Walker breakdown [43]. Remarkably, our results imply that measuring the chirality change of the domain wall would constitute a direct proof of the axial anomaly. Besides the electric-field-induced spin-torque, we also find that Fermi arc states bound to the domain wall naturally induce a hard-axis magnetic anisotropy. The hard-axis anisotropy is vital for domain wall motion without suffering an extremely early Walker breakdown. Hence, we find that high domain wall velocities are possible even if the intrinsic hard-axis anisotropy of the magnetic Weyl semimetal is small.

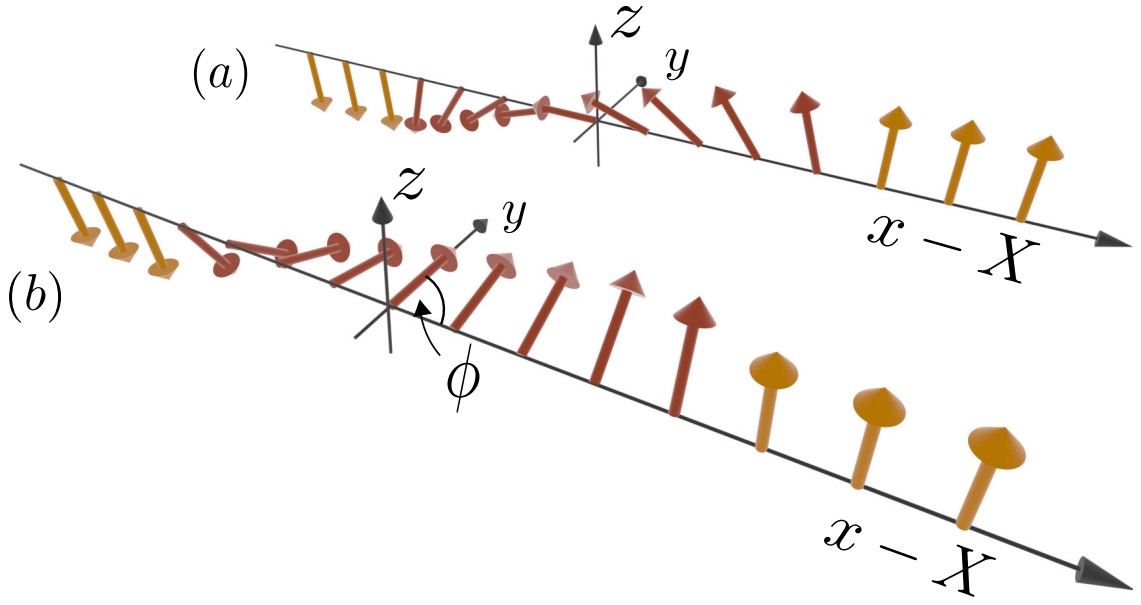

Figure 1: Examples of domain wall configurations with the domain wall plane in the $yz$-plane. Here $(a)$ is a Néel wall with internal angle $\phi = \pi$, and $(b)$ is a Bloch wall with internal angle $\phi = \pi/2$.

## 2 Background

For the purpose of making this work self-contained and to define notation, we first provide a short summary of the known physics of magnetic domain walls driven by external fields, as well as that of the axial anomaly and its connection to Weyl semimetals.

### 2.1 Field driven domain wall dynamics and the collective coordinate description

We consider a continuous domain wall in a ferromagnet, separating two domains for which the magnetization points in the $\pm z$ directions, and with domain wall plane in the $yz$-plane, see Fig. 1. It is convenient to describe the domain wall in terms of two coordinates, namely the center of the domain wall $X$, and the mean angle $\phi$ between the magnetization and the $xz$-plane, averaged over the domain wall width [44]. In terms of these coordinates the domain wall can be described by the unit magnetization:

$$\boldsymbol{m} = \left[ \frac{\cos \phi}{\cosh\left(\frac{x-X}{\lambda}\right)}, \frac{\sin \phi}{\cosh\left(\frac{x-X}{\lambda}\right)}, \tanh\left(\frac{x-X}{\lambda}\right) \right], \tag{6}$$

where $\lambda$ is the width of the domain wall. This expression makes clear that the value of $\phi$ describes the way the magnetization changes between the two domains. Therefore $\phi$ is often referred to as the internal angle of the domain wall. The case where $\phi = 0, \pi$ is titled a Néel wall [Fig. 1(a)] while a configuration with $\phi = \pm \pi/2$ is called a Bloch wall [Fig. 1(b)].

In the presence of an external magnetic field, the domain wall will generally move as it will be energetically favorable for the spins to align in a certain way with the magnetic field. For the configuration above, an external field along the $z$-axis induces domain-wall movement along the $x$-axis. While at small fields the domain wall moves rigidly, a sufficiently large magnetic field can further induce a rotation of the internal angle [45]. This movement of the domain wall can still be described by the two coordinates, by promoting $X$ and $\phi$ to the

dynamic collective coordinates $X(t)$ and $\phi(t)$, which are zero modes of fluctuations around the classical solutions [44]. The description in terms of $X(t)$ and $\phi(t)$ is valid for a rigid domain wall, which has translational invariance in the $x$ direction and rotational invariance around the $z$-axis.

The details of the domain wall dynamics can be expressed in the language of Lagrangian mechanics, through the ferromagnetic Lagrangian [45],

$$L_{\text{FM}} = \frac{\hbar}{a^3} \int \mathrm{d}^3x \; \dot\phi \, (\cos\theta - 1) - H_{\text{H}} - H_Z, \tag{7}$$

where the first term is the Berry phase term [46] with $\theta = 2\arctan\exp[-(x - X(t))/\lambda]$ and $a$ is a lattice constant. The second term is the Heisenberg Hamiltonian in the continuous limit [46]

$$H_{\text{H}} = \frac{1}{2a^3} \int \mathrm{d}^3x \; \left( Ja^2|\nabla \boldsymbol{m}|^2 - Km_z^2 + K_\perp m_y^2 \right), \tag{8}$$

where $J$ is a positive exchange constant. $H_{\text{H}}$ describes a magnet with an easy-axis in the $z$ direction, with easy-axis anisotropy $K$, and a hard-axis in the $y$ direction, with hard-axis anisotropy $K_\perp$. The domain wall width is $\lambda = \sqrt{Ja^2/K}$, and the hard-axis anisotropy is directly connected to the velocity of the domain wall, where we for simplicity choose the hard-axis anisotropy in the $y$ direction as this is the relevant [2] direction when we introduce the domain wall in a Weyl semimetal in Sec. 3. The external magnetic field $\boldsymbol{B}$ is included as a Zeeman coupling,

$$H_Z = \hbar/a^3 \int \mathrm{d}^3x \; \boldsymbol{m} \cdot \gamma \boldsymbol{B}, \tag{9}$$

where $\gamma$ is the electron gyromagnetic ratio. We are keeping the lattice spacing explicit in the Lagrangian, Eq. (7), rather than including it in the coupling constants, as we wish to write our expressions in terms of the unit magnetization which is proportional to the Weyl node separation, as we will see. A description in terms of $X(t)$ and $\phi(t)$ is valid despite the presence of anisotropy, as long as $K_\perp \ll K$ [45]. In terms of the domain wall solution, the Lagrangian is

$$L_{\text{FM}} = -\frac{2\hbar A}{a^3} \left[ \dot\phi X + v_\perp \sin^2\phi + \frac{\pi\lambda\gamma}{2} \left( B_x \cos\phi + B_y \sin\phi \right) - \gamma B_z X \right], \tag{10}$$

where $A$ is the cross section area of the domain, and $v_\perp = \lambda K_\perp/(2\hbar)$.

Only $B_z$ affects the translational motion of the domain wall as it couples to $X(t)$, so for simplicity we set the other components of the magnetic field to zero and let $\boldsymbol{B} = B\hat{z}$. The equations of motion of (10) are given by the Landau-Lifshitz-Gilbert equations in terms of the collective coordinates

$$\dot\phi + \frac{\alpha}{\lambda}\dot X = \gamma B, \tag{11}$$

$$\dot X - \alpha\lambda\dot\phi = v_\perp \sin 2\phi + \frac{a^3}{2\hbar A} T_e, \tag{12}$$

where the Gilbert damping parameter $\alpha$, which we take to be constant, enters via damping induced by a dissipation function $W = -\hbar A\lambda\alpha/2[(\dot X/\lambda)^2 + \dot\phi^2]$ [47]. For completeness and following [6], we have included a possible spin torque term $T_e$. In what follows we will focus on the magnetic driven doman wall dynamics, $T_e = 0$. The Landau-Lifshitz-Gilbert equations reduce to

$$\dot\phi = a_1 - a_2 \sin(2\phi), \tag{13}$$

---

[2] We show in section 3 that quantum fluctuations of Weyl fermions will induce a hard axis anisotropy in the $y$ direction.

where the constants $a_1 = \gamma B/(1 + \alpha^2)$ and $a_2 = \alpha \nu_\perp/[(\alpha^2 + 1)\lambda]$. The solution of Eq. (13) depends on the magnitude of the magnetic field. For an initial condition $\phi(0) = 0$ and for $B$ smaller than a a critical value

$$B_c = \frac{\alpha K_\perp}{2\hbar\gamma}, \tag{14}$$

for which $|a_1| = a_2$, $\phi$ is a constant in the long time limit:

$$\phi = \frac{1}{2}\arcsin\left(\frac{B}{B_c}\right). \tag{15}$$

In this limit the time averaged domain wall velocity is constant and proportional to the magnetic field, $\dot{X} = \lambda\gamma B/\alpha$. In the opposite limit where $B > B_c$, the solution for $\phi$ in Eq. (13) is oscillating and is no longer a constant:

$$\phi = \frac{\arctan\left[a_1\tan\left(\sqrt{a_1^2 - a_2^2}\,t\right)\right]}{\sqrt{a_1^2 - a_2^2} + a_2\tan\left(\sqrt{a_1^2 - a_2^2}\,t\right)}, \tag{16}$$

and the time averaged domain wall velocity

$$\langle\dot{X}\rangle = \frac{\lambda\gamma}{\alpha}B\left[1 - \frac{1}{1 + \alpha^2}\sqrt{1 - \left(\frac{B_c}{B}\right)^2}\right], \tag{17}$$

begins at first to decrease once the magnetic field becomes larger than $B_c$, to then again grow linearly with the magnetic field as $B \gg B_c$. The transition between the two solutions is called Walker breakdown [43] and coincides with the maximum velocity of the domain wall for the regime of constant $\phi$ solutions.

## 2.2 The axial anomaly in Weyl semimetals

Weyl semimetals are three dimensional semimetals which at low energies are described by quasi-particles which are chiral fermions [48]. These are massless fermions with a given (right or left) handedness. Their dispersion is conical and crosses through a node called a Weyl point, with Weyl points always coming in pairs of opposite chirality [48–50]. The Weyl points in a Weyl semimetal are separated in either energy or momentum, which, depending on the system and the number of Weyl points, breaks either inversion symmetry, time-reversal symmetry or both. We consider only a time-reversal-breaking Weyl semimetal consisting of two Weyl nodes separated in momentum space. This is a magnetic Weyl semimetal where the momentum space separation is given by a magnetization vector, as described by the Hamiltonian in Eq. (1).

Due to the presence of chiral fermions, Weyl semimetals exhibit the axial anomaly: the non-conservation of axial charge Eq. (5)

$$\partial_t n_5 = \frac{e^2}{2\hbar^2\pi^2}\boldsymbol{E}\cdot\boldsymbol{B} - \frac{n_5}{\tau}, \tag{18}$$

where the axial number density $n_5 = n_R - n_L$ is the difference in the number of right and left handed fermions. The last term models scattering between the two Weyl cones in a relaxation-time approximation with inter-node scattering time $\tau$. Parallel magnetic and electric fields result in a depletion of left-handed fermions which are transmuted into right-handed fermions, or vice versa depending on the mutual sign of the electric and magnetic fields. The transfer of fermions of one chirality into the other, generates a difference of chemical potential between the two Weyl cones; this difference is called the axial chemical potential, $\mu_5 = (\mu_R - \mu_L)/2$.

The axial chemical potential is conveniently expressed in terms of the axial number density, which in turn is obtained from the anomaly equation (18), which for constant electromagnetic fields results in a steady state solution of the axial density:

$$n_5 = \frac{e^2 \tau}{2\hbar^2 \pi^2} \boldsymbol{E} \cdot \boldsymbol{B}. \tag{19}$$

In the limit of a small magnetic field, $\hbar e B \ll \mu_5^2 / v_F^2$ and a small axial chemical potential, $\mu_5 \ll \mu, k_B T$, where $\mu = (\mu_{\mathrm{L}} + \mu_{\mathrm{R}})/2$ is the average chemical potential, $T$ the temperature and $k_B$ the Boltzmann constant, the axial chemical potential [38]

$$\mu_5 = \frac{3\hbar v_F^3 e^2 \tau}{2(\pi^2 k_B^2 T^2 + 3\mu^2)} \boldsymbol{E} \cdot \boldsymbol{B} \tag{20}$$

is linear in $n_5$.

## 3 Model and effective Lagrangian

We now turn to the main objective of this work, combining domain wall physics with Weyl fermions and anomalies. For this end we consider a Weyl semimetal which hosts a domain wall in the magnetization as described by Eq. (6), and are ultimately interested in the dynamics of this domain wall under the influence of external magnetic and electric fields. This requires a collective coordinate description of the domain wall that further includes the coupling of the magnetization with the Weyl fermions. In this section we present and discuss the effective Lagrangian for the magnetization under the influence of external electromagnetic fields, in collective coordinates, obtained by integrating out the fermionic degrees of freedom; for clarity many details of the derivation are relegated to appendices.

The initial description of the system is given by a Lagrangian with two contributions, one for the ferromagnet and the other one for the Weyl fermions and their coupling to magnetization,

$$L_{\mathrm{tot}} = L_{\mathrm{FM}} + L_{\mathrm{Weyl}}. \tag{21}$$

The ferromagnet Lagrangian is

$$L_{\mathrm{FM}} = -\frac{2\hbar A}{a^3} \left[ \dot{\phi} X + \frac{\pi \lambda \gamma}{2} \left( B_x \cos\phi + B_y \sin\phi \right) - \gamma B_z X \right], \tag{22}$$

where we at this point do not include any intrinsic hard-axis anistropy but, as we will see, it will be induced by the coupling to the Weyl fermions. The Weyl Lagrangian

$$L_{\mathrm{Weyl}} = \int \mathrm{d}^3 x \left( \bar{\Psi} i \gamma^0 \partial_0 \Psi - H \right), \tag{23}$$

$$H = \bar{\Psi} \left( -i v_F \gamma^i \partial_i + e \gamma^\mu A_\mu - e v_F b_\mu \gamma^\mu \gamma^5 \right) \Psi, \tag{24}$$

is the low energy description of the Weyl semimetal in units with $\hbar = c = 1$. Here the Greek indices correspond to the four-dimensional space-time coordinates $t, x, y, z$ and Latin indices to space coordinates $x, y, z$. Repeated indices are summed over. The gamma-matrices, $\gamma^\mu$, obey the anti-commutation relations $\{\gamma^\mu, \gamma^\nu\} = 2\eta^{\mu\nu}$, where our chosen metric signature is $\eta^{\mu\nu} = \mathrm{diag}[1, -1, -1, -1]$. The fifth gamma-matrix is defined as $\gamma^5 = i\gamma^0 \gamma^1 \gamma^2 \gamma^3$, and it couples with opposite sign to Weyl fermions of opposite chirality. We work in the Weyl representation, where $\gamma^5 = -\tau_z \otimes \mathbb{1}$, and Eq. (24) matches Eq. (1). $L_{\mathrm{Weyl}}$ couples an external vector

potential $A_\mu$ through minimal coupling, where $e$ is the elementary charge. To simplify the notation in Eq. (24), we have re-scaled the vector potential to include the velocity, $A_\mu = (A_t, v_F A_i)$. The magnetization $b_\mu = (0, \boldsymbol{b})$ is purely space-like and is given by $\boldsymbol{b}(x, t) = \Delta/(e v_F)\boldsymbol{m}(x, t)$, with $\Delta$ an exchange coupling between the electrons and the magnetization. We consider the magnetization to constitute a background magnetization in the $z$ direction with fluctuations on top, and expand the effective theory of the Weyl semimetal in terms of these fluctuations. The background magnetization is along the $z$ direction, $\tilde{b}_i = \Delta/(e v_F)(0, 0, \tanh[x/\lambda])$, and is defined to be the zeroth order term in an expansion of $b_z$ around $x = X(t)$. The fluctuations in the $z$ direction, $\delta b_z$, contain all higher order terms in the expansion of $b_z$, and the fluctuations in the $x$ and $y$ directions are $\delta b_x = m_x$ and $\delta b_y = m_y$. Considering these definitions, the total magnetization is the sum containing the background and the fluctuations, $b_i = \tilde{b}_i + \delta b_i$. The magnetization couples to $\gamma^5$ in Eq. (23) and therefore distinguishes between the chirality of the fermions.

The domain wall gives rise to two types of fermionic solutions: bound states confined to the domain wall, and extended plane-wave bulk states away from the domain wall. The spectrum of the bound states exhibits two chiral zero modes of opposite chirality, these are the Fermi arcs [51]. Both Fermi arcs follow the same linear dispersion relation, $E = v_F k_y$ [52].

By integrating out the fermionic degrees of freedom separately considering the bound-state and extended-state solutions, we obtain an effective Lagrangian for the magnetization

$$L_{\text{eff}} = L_{\text{bound}} + L_{\mu_5}, \tag{25}$$

which takes into account terms up to and including second order in the magnetization fluctuations. The contribution from the Fermi arc states is

$$L_{\text{bound}} = A\lambda K_\perp^{\text{eff}} \sin^2 \phi. \tag{26}$$

Note that $m_y \propto \sin \phi$, so $L_{\text{bound}}$ is an effective hard-axis anisotropy, where $K_\perp^{\text{eff}} = \Delta^2/(L_z \lambda h v_F)$ is the effective hard-axis anisotropy per unit volume, and $L_z$ is the sample thickness in the $z$ direction. The form of the induced anisotropy is model-dependent as it directly depends on the Fermi arc states (see App. A.1); here it couples to $m_y$ only since in our model the Fermi arcs only disperse in the $k_y$ direction. For a detailed derivation of $L_{\text{bound}}$ see App. A.1.

The derivation of the second contribution to the effective Lagrangian, $L_{\mu_5}$, differs from the derivation of $L_{\text{bound}}$; rather than starting from the Weyl Lagrangian in Eq. (23), the existence and specific form of $L_{\mu_5}$ are instead derived from the axial separation effect. This effect gives an axial current proportional to the axial chemical potential and an axial magnetic field [35–37]

$$J_5^i = \frac{e^2}{6\pi^2 \hbar^2} \mu_5 B_5^i. \tag{27}$$

The correct coefficient for this current can be obtained by noticing that the corresponding Feynman diagram contains three insertions of axial fields ($\mu_5$ is a component of an axial field $A_t^5$), which means it has three insertions of $\gamma^5$ matrices. Due to $(\gamma^5)^3 = \gamma^5$, the structure of the Feynman integral for the axial separation effect is analogous to that for the chiral magnetic effect, which has two insertions of $A_\mu$ and only one insertion of $A_\mu^5$. They differ however by a relative factor of $1/3$ [28]: since the diagram for the axial separation effect has three insertions of the same field $A_\mu^5$, it is accompanied by a factor of $1/3!$ accounting for permutations, while the chiral magnetic effect has two insertions of the same field $A_\mu$, and therefore goes with a factor $1/2!$. $L_{\mu_5}$ can be obtained from the definition of the current: $J_5^i = \delta L_{\mu_5}/\delta A_i^5$, so that $L_{\mu_5} = \frac{e^2}{6\pi^2 \hbar^2} \mu_5 \int d^3 x \, \epsilon^{ijk} A_i^5 \partial_j A_k^5$. In terms of the magnetization, which couples as an axial field $A_5 = \Delta \boldsymbol{m}/e v_F$, we finally obtain

$$L_{\mu_5} = \frac{\Delta^2}{12\pi^2 v_F^2 \hbar^2} \mu_5 \int \mathrm{d}^3 x \, \varepsilon^{ijk} m_i \partial_j m_k. \tag{28}$$

One can understand $L_{\mu_5}$ in terms of a spin-torque exerted on the magnetization, as explained in the introduction. More precisely, the presence of a finite axial current $J_5$ influences the magnetization dynamics through a torque $T_e \propto m \times J_5$, see Eq. (3). Being proportional to $\mu_5$, $L_{\mu_5}$ only emerges once the axial chemical potential is dynamically generated, which is a result of the axial anomaly. When the expression for the axial chemical potential, Eq. (20), is inserted,

$$L_{\mu_5} = \frac{A\Delta^2 v_F e^2 \tau}{8\pi\hbar(\pi^2 k_B^2 T^2 + 3\mu^2)}(E \cdot B)\sin\phi. \tag{29}$$

The resulting spin torque, as appearing in Eq. (12), then reads

$$T_e = -\frac{A\Delta^2 v_F e^2 \tau}{8\pi\hbar(\pi^2 k_B^2 T^2 + 3\mu^2)}(E \cdot B)\cos\phi. \tag{30}$$

There still exist two further contributions to the effective Langrangian. One comes from the chiral separation effect [35–37], $J_5 \propto \mu B$, which gives a term in the Lagrangian $\propto \mu A^5 \cdot B \propto m \cdot B$. This term just renormalizes the Zeeman coupling to the magnetization, so nothing qualitative comes from it and we neglect it. The other term has the form (see App. A.2)

$$L_{CS}^5 = \frac{A\Delta^3}{9\pi^2 v_F^3 \hbar^2}\dot\phi X. \tag{31}$$

Again, this term only renormalizes the Berry phase term in the Heisenberg Hamiltonian in Eq. (22). It is small, in terms of typical parameter values [3], in comparison to the Berry phase term, and we neglect it.

The final expression for the effective Lagrangian for the magnetization is the sum of the ferromagnet contribution Eq. (22), the effective contribution from the Fermi-arcs, Eq. (26), and the bulk term arising from the axial anomaly, Eq. (29):

$$L_M = -\frac{2\hbar A}{a^3}\left[\dot\phi X + \frac{\pi\lambda\gamma}{2}\left(B_x\cos\phi + B_y\sin\phi\right) - \gamma B_z X\right] - C_1\sin^2\phi + C_2(E \cdot B)\sin\phi, \tag{32}$$

where the coefficients are

$$C_1 = A\lambda K_\perp^{\text{eff}}, \tag{33}$$

$$C_2 = \frac{A\Delta^2 v_F e^2 \tau}{8\pi\hbar(\pi^2 k_B^2 T^2 + 3\mu^2)}. \tag{34}$$

The Weyl physics, through the interaction between the Weyl fermions and the magnetization, is thus responsible for inducing a hard-axis anisotropy and an electrically modulated spin-torque. This is one of the main results of this work.

As a final comment, we note that we do not capture the term in Eq. (4) from Ref. [17]. That term is a Fermi surface term, and the chemical potential needs to be explicitly introduced in the calculations to capture it. Our focus in the present work is on the contrary on the physics of the axial anomaly.

---

[3]Lattice constant $a = 0.5$ nm, domain wall width $\lambda = 10$ nm, domain length in $y$ direction $L_y = 10\mu$m, thickness in $z$ direction $L_z = 3$nm, Fermi velocity $v_F = 5 \cdot 10^5$ m/s, inter-valley scattering rate $\tau = 1$ ns, Gilbert damping constant $\alpha = 0.01$, half the length of the Weyl node separation $k_\Delta = |b| = 0.2\pi/a$, exchange energy $\Delta = k_\Delta v_F \hbar$, chemical potential $\mu = 10$meV, temperature $T = 300$K.

# 4 Axial anomaly manipulation of the domain wall

With the expression for the effective Lagrangian, Eq. (32), at hand, we now proceed to show how the interaction between the Weyl fermions and the magnetization induces a means to control the equilibrium value of the internal angle through an electric field. Specifically, we demonstrate that due to the axial anomaly induced spin-torque, Eq. (30), the chirality of the domain wall changes smoothly as a function of the electric field. By a chiral domain wall we refer to a Bloch wall where the two chiralities corresponds to the internal angles $\phi = \pm\pi/2$ [53]. Intuitively, the effect can be understood by noticing that the axial anomaly induced term $L_{\mu_5}$ resembles a Dzyaloshinskii-Moriya interaction, $L_{\mu_5} \propto \boldsymbol{m} \cdot \nabla \times \boldsymbol{m}$. Such terms, that usually arise when inversion symmetry is broken, are known to influence the internal structure or chirality of domain walls [54, 55].

We also display how the dynamics of the domain wall depend on the induced hard-axis anisotropy, Eq. (26), and how the electric field, through the spin-torque due to Eq. (30), delays the onset of a Walker break down.

## 4.1 Equilibrium configuration of the domain wall

The equilibrium configuration of the domain wall is defined as the value of the internal angle for which the potential energy of the domain wall is minimal. This angle depends on the electric and magnetic fields, and without loss of generality we consider magnetic fields with no $z$ component, as $B_z$ only couples to $X(t)$ in the Lagrangian Eq. (32), and therefore does not affect the $\phi$ dependence of the potential energy. We separately consider the magnetic field configurations $\boldsymbol{B} = B_x \hat{x}$ and $\boldsymbol{B} = B_y \hat{y}$, to clarify the influence of the electric field on the equilibrium configuration. When there are no external fields, the equilibrium configuration of the domain wall is a Néel wall, with degenerate potential minima for $\phi_0 = 0, \pi$.

For a magnetic field $\boldsymbol{B} = B_x \hat{x}$, and a zero electric field the potential energy in Eq. (32) is

$$V_{B_x}(\phi) = C_1 \sin^2 \phi + C_3 B_x \cos \phi, \tag{35}$$

$$C_3 = \frac{\pi\hbar\lambda A\gamma}{a^3}, \tag{36}$$

which has a minima at,

$$\phi_0 = \begin{cases} 0 & B_x < 0, \\ \pi & B_x > 0. \end{cases} \tag{37}$$

The equilibrium configuration is therefore a Néel wall for all magnetic fields, sharply changing the chirality as the magnetic field changes sign. In an electric field $\boldsymbol{E} = E_x \hat{x}$ parallel to the magnetic field,

$$V_{E_x \cdot B_x}(\phi) = C_1 \sin^2 \phi + C_3 B_x \cos \phi - C_2 (E_x \cdot B_x) \sin \phi, \tag{38}$$

for which the minima smoothly varies as a function of the electric field. We define a critical electric field $E_c = C_3/C_2$, which for typical parameter values [3] is $E_c = 39\,\text{kV/m}$. For a negative magnetic field and $E_x > 0$, the angle $\phi_0$ decreases as the electric field increases, and approaches a Bloch configuration $\phi_0 = -\pi/2$ as $E_x \gg E_c$. When $E_x < 0$, $\phi_0$ instead increases towards $\phi_0 = \pi/2$ with decreasing electric field. In Fig. 2 $\phi_0$ is plotted against $E_x$ for three different values of the magnetic field, showing that the larger the magnitude of the magnetic field is, the smaller the electric field required to reach a Bloch configuration.

For a magnetic field $\boldsymbol{B} = B_y \hat{y}$, the potential energy

$$V_{B_y}(\phi) = C_1 \sin^2 \phi + C_3 B_y \sin \phi \tag{39}$$

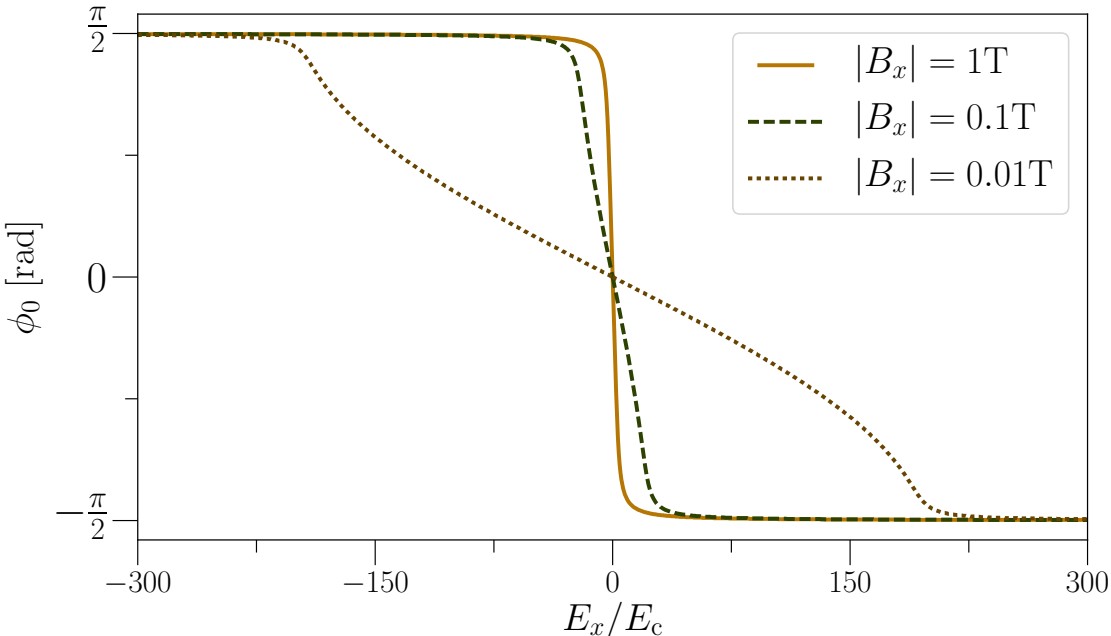

Figure 2: The internal angle, $\phi_0$ as a function of the electric field $E_x$, for three different values of $B_x$, and where $B_y = B_z = 0$, and $B_x < 0$ for parameter values [3]. The solid line corresponds to $|B_x| = 1$ T, the dashed —— line corresponds to $|B_x| = 0.1$T and the dotted .. line to $|B_x| = 0.01$T.

has a minima at

$$\phi_0 = -\arcsin\left(\frac{C_3 B_y}{2C_1}\right), \tag{40}$$

which for $B_y \in [-B_{c,y}, B_{c,y}]$, with $B_{c,y} = 2C_1/C_3$, assumes values in the open interval $\phi_0 \in (-\pi/2, \pi/2)$. When $B_y < -B_{c,y}$ ($B_y > B_{c,y}$) the domain wall is a Bloch wall with internal angle, $\phi_0 = \pi/2$ ($\phi_0 = -\pi/2$). For $\boldsymbol{E} = E_y \hat{y}$, the mimima of the potential energy

$$V_{E_y \cdot B_y}(\phi) = C_1 \sin^2 \phi - \left[C_2\left(E_y \cdot B_y\right) - C_3 B_y\right]\sin\phi \tag{41}$$

depends on the electric field:

$$\phi_0 = \arcsin\left[\frac{C_2\left(E_y \cdot B_y\right) - C_3 B_y}{2C_1}\right]. \tag{42}$$

This minima corresponds to the Néel configuration when the argument is zero, at the critical value $E_y = E_c = C_3/C_2$. By varying the electric field, the Néel configuration smoothly changes into a Bloch configuration, $\phi_0 = \tilde{q}\,\pi/2$, where the chirality of the wall, $\tilde{q} = q\,\text{sgn}(B_y)$, $q = \pm 1$, depends on the sign of the magnetic field. The Bloch configuration is reached once the electric field is $E_y = E_c + \tilde{q}\,2C_1/(|B_y|C_2)$. The internal angle, Eq. (42), is depicted in Fig. 3 as a function of the electric field for three different values of the magnetic field $B_y < 0$.

The results in this section can be summarized as follows: For electric and magnetic fields in the direction ($\hat{x}$) perpendicular to the domain wall plane, the equilibrium configuration changes smoothly between a Néel wall at $E_x = 0$, to approach a Bloch wall as the magnitude of the electric field increases (Fig. 2). For fields lying in the domain wall plane and perpendicular to the easy axis (pointing along $\hat{y}$) the equilibrium configuration at $E_y = 0$ changes between Néel and Bloch configurations as the magnetic field is varied. For nonzero $E_y$ the

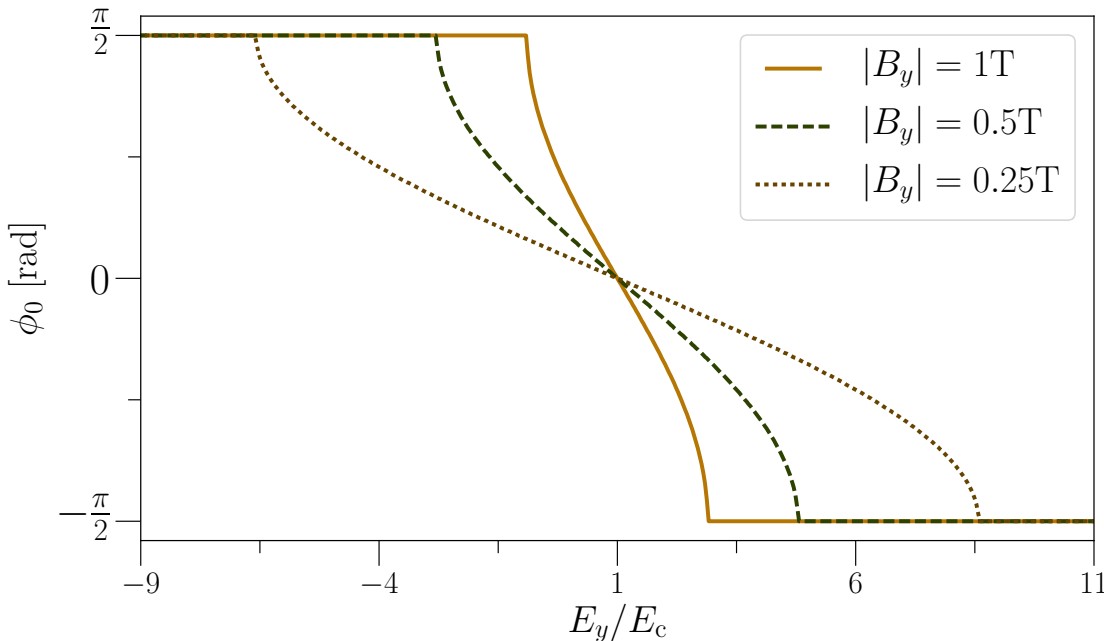

Figure 3: The internal angle, $\phi_0$ as a function of the electric field $E_y/E_c$, for three different values of $B_y$, and where $B_x = B_z = 0$, and $B_y < 0$ for parameter values [3]. The solid line corresponds to $|B_y| = 1$ T, the dashed line corresponds to $|B_y| = 0.5$T and the dotted line to $|B_y| = 0.25$T.

same result is obtained by keeping the magnetic field constant and varying the electric field (Fig. 3). The strength of the electric field needed to approach a Bloch configuration is in both cases determined by the magnitude of the magnetic field, the stronger the magnetic field the smaller the electric field has to be to reach a Bloch configuration. The interaction between the Weyl fermions and the magnetization thus induces a coupling of the magnetization to the electric field which allows for the electric field to be used to flip the chirality of a Bloch wall. Since this interaction is due to the axial anomaly, an observation of the chirality flip serves as an indirect measurement of the axial anomaly itself.

## 4.2 Maximum domain wall velocity

To move the domain wall along the x-axis a magnetic field must have a component along the easy axis, the $z$-direction, as only this component couples to $X(t)$ in Eq. (32). For simplicity we therefore choose the magnetic field to be $\boldsymbol{B} = B\hat{z}$. The equations of motion of the Lagrangian describing the magnetization in Eq. (32),

$$\dot{\phi} + \frac{\alpha}{\lambda}\dot{X} = \gamma B \tag{43}$$

$$\frac{2A\hbar}{a^3}\dot{X} - C_1 \sin 2\phi - \frac{2A\hbar\lambda\alpha}{a^3}\dot{\phi} = -C_2 E_z B_z \cos\phi, \tag{44}$$

contain two contributions due to the coupling between the electrons and the magnetization. These contributions, as we now show, affect the occurrence of the Walker breakdown, and in turn the maximal velocity $\dot{X}_{\mathrm{max}} = \lambda\gamma B_c/\alpha$ of the domain wall in the $\dot{\phi} = 0$ regime; here, as in Sec. 2.1, $|B_c|$ is the magnitude of the magnetic field at which $\dot{\phi}$ becomes nonzero.

For a zero electric field, Eqs. (43) and (44) reduce to

$$\dot{\phi} = a_1 - a_2 \sin(2\phi), \tag{45}$$

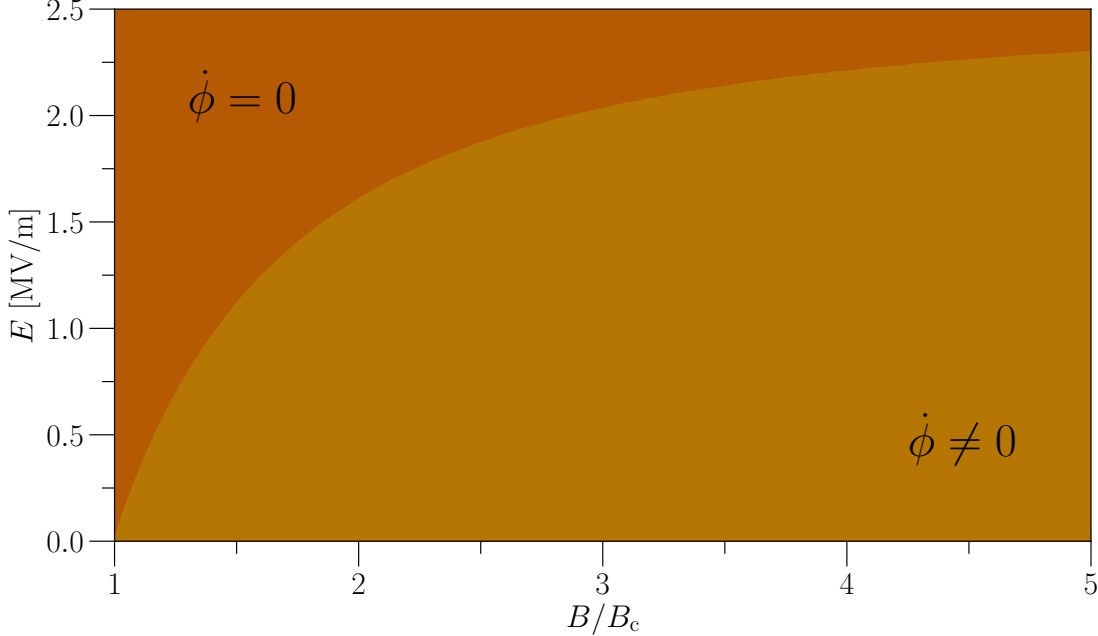

Figure 4: The electric, $E_z$, and magnetic, $B_z/B_c$, field dependence of the boundary between the Walker regime, $\dot{\phi} = 0$ and the regime where the internal angle rotates, $\dot{\phi} \neq 0$ for parameter values [3].

where the constants $a_1 = \gamma B/(1 + \alpha^2)$ and $a_2 = \alpha a^3 K_\perp^{\text{eff}}/(2\hbar[1 + \alpha^2])$. The Walker breakdown occurs when $|a_1| = a_2$ at a critical magnetic field $B_c = a_2(1 + \alpha^2)/\gamma$, which for physical parameter values [3] is $B_c = 15\,$mT. The corresponding velocity is $\dot{X}_{\text{max}} = 2617\,$m/s, which is remarkable: This means that the DW in the Weyl semimetal can move at high velocities even if the intrinsic hard-axis anisotropy is vanishingly small (we have taken $K_\perp = 0$). As a comparison, the maximum velocity in nanowires are of the order of 100 m/s [56, 57]; the hard axis anisotropy for permalloy nanowires range between $K_\perp = k_B(0.1 - 1)$ K [45], which for physical parameter values [3] yields a maximum velocity of approximately $65 - 650$ m/s.

For an electric field parallel to the magnetic field, the equations of motions in Eq. (43) and Eq. (44) combine into

$$\dot{\phi} = a_1 - a_2 \sin(2\phi) + a_3 E_z \cos\phi, \tag{46}$$

where $a_3 = \alpha a^3 C_2 B/[2\hbar A\lambda(1 + \alpha^2)]$. The existence of the electric field delays the onset of the Walker breakdown, as displayed in Fig. 4, depicting how the boundary between the Walker regime, $\dot{\phi} = 0$, and beyond, $\dot{\phi} \neq 0$, depends on the electromagnetic fields. To estimate the field strengths at which Walker breakdown takes place we note that $F(\phi) = -a_2 \sin(2\phi) + a_3 E_z \cos\phi$ being a sum of trigonometric functions of $\phi$, is bounded both from above and below. If the coefficient $a_1 > |\max[F(\phi)]|$, then the right hand side can never become zero, and therefore there is no solution in which the domain wall angle does not rotate. Therefore, the condition $a_1 = |\max[F(\phi)]|$ gives a strict bound on the when the Walker breakdown happens. Note that this does not rigorously exclude solutions with $\dot{\phi} \neq 0$ at lower values of the magnetic field. In the absence of an electric field, this condition is however equivalent to the relation $B = B_c$ derived above.

## 5 Discussion

We have shown how the joint action of two effects, the axial anomaly and the axial separation effect, induces an electric-field-modulated spin torque in magnetic Weyl semimetals. We have demonstrated that the presence of this spin torque allows for the control of the chirality of domain walls, which can be flipped in a controllable manner by a varying electric field. For electric and magnetic fields in the $x$ direction and for $|B_x| \sim 1\,\text{T}$, the chirality of the Bloch wall is flipped from $\pi/2$ to $-\pi/2$ by increasing the electric field from $E_x = 0$ to $|E_x| \sim 2.7\,\text{MV/m}$ for typical parameter values [3]. The same chirality flip is obtained for electric and magnetic fields in the $y$ direction. In this case, for $B_y \sim -1\,\text{T}$ the electric field needed to reach a Bloch wall with negative chirality is $E_y = 112\,\text{kV/m}$, and to reach Bloch wall with positive chirality is $E_y = -35\,\text{kV/m}$. These results could serve for developing logic gate designs based on spin wave technology [40–42], and, perhaps more striking, the experimental observation of such electric field mediated chirality changes would constitute a direct signature of the axial anomaly. Furthermore, the electric-field-induced spin torque can also be used to electrically delay the onset of Walker breakdown in the domain wall dynamics. This, in addition with an effectively induced magnetic anisotropy by the Fermic arc states bound to the domain wall, permits high domain wall velocities even if the intrinsic hard-axis anisotropy of the Weyl semimetals is small. For typical parameter values [3], and assuming a vanishing intrinsic hard-axis anisotropy, the maximum time averaged domain wall velocity in the Walker regime at zero electric field is $\dot{X}_{\max} = 2.6\,\text{km/s}$. This value can be further increased by a factor of 3 by applying an electric field of $E = 2\,\text{MV/m}$.

## Acknowledgments

This project has received funding from the European Research Council (ERC) under the European Union's Horizon 2020 research and innovation programme (grant agreement No.679722). Y.F. acknowledges financial support through the Programa de Atracción de Talento de la Comunidad de Madrid, Grant No. 2018-T2/IND-11088. A.C. acknowledges financial support through European Union structural funds, the Comunidad Autonoma de Madrid (CAM) NMAT 2D-CM Program (S2018-NMT-4511) and the Ramon y Cajal program through the grant RYC20 18-023938-I.

## A  Effective action for the magnetization

In this section we obtain the effective action for the magnetization due to its coupling to the fermions. The starting point is the Weyl semimetal action, which at low energies is

$$S_{\text{Weyl}} = \int \mathrm{d}^4 x \left( \bar{\Psi} i \gamma^0 \partial_0 \Psi - H \right), \tag{47}$$

$$H = \bar{\Psi} \left( -i\, v_F \gamma^i \partial_i + e \gamma^\mu A_\mu - e\, v_F\, b_\mu \gamma^\mu \gamma^5 \right) \Psi, \tag{48}$$

where for notational convenience we define $\partial_\mu = (\partial_0, v_F \partial_i)$ and $A_\mu = (\partial_0, v_F A_i)$, where the the gauge potential is partially fixed through $A_\mu \to A_\mu(t, y)$. Note that the chemical potential is zero, $\mu = 0$, in Eq. (48), as the relevant physics of the axial anomaly and its consequences is captured without the need to introduce a finite chemical potential. We decompose the total magnetization into a sum $b_i = \tilde{b}_i + \delta b_i$, of the background magnetization, $\tilde{b}_i = \Delta/(e\,v_F)(0, 0, \tanh[x/\lambda])$ and fluctuations $\delta b_i = \Delta/(e\,v_F)(m_x, m_y, \delta m_z)$. The domain

wall generates two types of solutions, Fermi arc states bound to the domain wall, and extended bulk states. The fermionic states are integrated out to obtain an effective theory for the magnetization, giving two separate terms for the two solutions. These are then further expanded in both the fluctuations $\delta b_i$ and the gauge fields $A_\mu$, up to and including quadratic order in the fluctuations of the magnetization. We employ the chiral basis of the gamma matrices as it decouples the chiralities:

$$\gamma^0 = \begin{pmatrix} 0 & \mathbb{1} \\ \mathbb{1} & 0 \end{pmatrix}, \; \gamma^i = \begin{pmatrix} 0 & \sigma^i \\ -\sigma^i & 0 \end{pmatrix}, \; \gamma^5 = \begin{pmatrix} -\mathbb{1} & 0 \\ 0 & \mathbb{1} \end{pmatrix}, \tag{49}$$

where $\mathbb{1}$ is the identity matrix in two dimensions, and $\sigma^i$ are the Pauli matrices.

We now turn to the details of this derivation separately for the two contributions from Fermi arcs and bulk states in the next two subsection, where we only consider the parity odd bulk terms, as these are the most important for our discussion. In the last subsection we comment on the effect of the parity even terms, which result only in renormalization of coupling constants.

### A.1 The effective action due to the bound states

The momentum space Hamiltonian for $A_\mu = \delta b_\mu = 0$, corresponding to the Weyl action in Eq. (47) is

$$\hat{H}_0 = v_F \left[ (\tau_z \otimes \boldsymbol{\sigma}) \cdot \boldsymbol{k} - e(\mathbb{1} \otimes \boldsymbol{\sigma}) \cdot \tilde{\boldsymbol{b}} \right], \tag{50}$$

where $\boldsymbol{\sigma}$ and $\boldsymbol{\tau}$ act on the spin and chirality space respectively. This Hamiltonian is translational invariant in the $y$ and $z$ directions and the wave function in these directions is a plane wave, and the remaining $x$-dependence of the wave function is obtained by noting that the chiralities decouple in Eq. (50). The Fermi arc states, which are the zero mode solutions to Eq. (50) with dispersion $E = v_F k_y$ are [52]:

$$\Psi_{\text{L/R}}(\boldsymbol{x}) = \int \mathrm{d}k_z e^{ik_z z} \psi_{\text{L/R}}(x, y, k_z), \tag{51}$$

where the subscript refers to the left/right handedness ($\tau_z = \pm 1$, with a slight abuse of notation, using the same notation for the eigenvalues of $\tau_z$ as the matrix itself, the context should make clear which is considered) of the fermions:

$$\psi_{\text{L}}(x, y, k_z) = \frac{1}{\sqrt{2}} \begin{pmatrix} -i \\ 1 \end{pmatrix} \Phi_{\text{L}}(x, k_z) \varphi(y), \tag{52}$$

$$\psi_{\text{R}}(x, y, k_z) = \frac{1}{\sqrt{2}} \begin{pmatrix} 1 \\ -i \end{pmatrix} \Phi_{\text{R}}(x, k_z) \varphi(y). \tag{53}$$

Here $\varphi(y) = e^{ik_y y}/\sqrt{L_y}$, and the states are normalized such that $\int \mathrm{d}y |\varphi(y)^2| = 1$, and [52]

$$\Phi_{\text{L/R}}(x, k_z) = \begin{cases} N e^{-\tau_z k_z x - k_\Delta \lambda \ln(\cosh(x/\lambda))} & |k_z| < k_\Delta \\ 0 & |k_z| \geqslant k_\Delta, \end{cases} \tag{54}$$

where $2k_\Delta$ is the Weyl node separation. The constant $N$ is defined through the normalization of the complete wave function Eq. (51), giving the condition:

$$\int \mathrm{d}k \, \mathrm{d}x |\Phi_{\text{L/R}}(x, k_z)|^2 = 1. \tag{55}$$

Note that in addition to the Fermi arc states, there might be additional gapped bound states [52], depending on the width of the domain wall. The gapped bound states, however, are not

one dimensional, as they disperse in the other dimensions. In this sense, the physics of these bound states is similar to those of bulk states, and one can expect them to contribute by just renormalizing the terms of the effective action given by the bulk states. In other words, they are not going to give any new qualitative physics. This has been previously discussed in [12].

By inserting the Fermi arc solutions (51) into the Weyl action (47) we are left with the expression

$$S_{L/R} = \int dt\, dx\, dy \int dk_z\, \varphi^\dagger(t,y)\Phi_{L/R}(x,k_z)[iD_t - iD_y - \tau_z \Delta m_y]\varphi(t,y)\Phi_{L/R}(x,k_z), \quad (56)$$

for the action, where the operator $D_\mu = \partial_\mu + ieA_\mu$ and the time dependence has been included in the function $\varphi(t,y)$. We expand the fluctuation $m_y = \sin\phi / \cosh([x-X(t)]/\lambda)$ in Eq. (56) around the domain wall center $x - X(t) = 0$ and only keep the zeroth order term in this expansion, namely $m_y^0 = \sin\phi$. With $m_y^0$ independent of $x$, the integration over $x, k_z$ in Eq. (56) yields simply one. After integrating out the fermionic dependence the effective action, with respect to the gauge field and the magnetization, is

$$\Gamma_{\text{bound}}^{(\tau_z)} = -i \ln \det\left(iD_t - iD_y - \tau_z \Delta m_y^0\right). \quad (57)$$

Expanded to second order in the gauge field and magnetization, using a gauge invariant regularization, gives the expression [58]

$$\Gamma_{\text{bound}}^{(\tau_z,2)} = \frac{1}{2\pi v_F \hbar} \int dt\, dy\, a_\mu^{(\tau_z)}\left(\eta^{\mu\nu} - \frac{\partial^\mu \partial^\nu}{\partial^2} + \frac{\varepsilon^{\mu\alpha}\partial_\alpha \partial^\nu - \varepsilon^{\beta\nu}\partial^\mu \partial_\beta}{2\partial^2}\right)a_\nu^{(\tau_z)}, \quad (58)$$

where $a_\mu^{(\tau_z)} = (eA_0, ev_F A_y - \tau_z \Delta m_y^0)$. The full action is the sum over both chiralities, but since the magnetization couples with opposite sign for opposite chiralities, the terms which mix the magnetization and $A_\mu$ cancel, and the remaining effective action contains only two terms, one quadratic in the magnetization and one quadratic in the gauge field. The effective action for the magnetization reduces to [12,14]

$$\Gamma_{\text{bound}} = -\frac{\Delta^2}{2\pi v_F \hbar} \int dt\, dy\, m_y^0(t) m_y^0(t). \quad (59)$$

This takes the form of a Fermi-arc induced hard-axis anisotropy. We therefore define the effective hard-axis anisotropy,

$$K_\perp^{\text{eff}} = \frac{\Delta^2}{L_z \lambda h v_F}, \quad (60)$$

where $L_z$ is the thickness of the domain wall in the $z$ direction. The Lagrangian corresponding to the action in Eq. (59) for a sample width $L_y$ in the $y$ direction is then

$$L_{\text{bound}} = -L_y L_z \lambda\, K_\perp^{\text{eff}} \sin^2 \phi. \quad (61)$$

Note that the form of the effective Lagrangian is model dependent. The linearly dispersing (with $E \propto k_y$) Fermi arcs in our model are eigenfunctions of $\sigma_y$ within one valley, causing the action in Eq. (56) to only depend on the magnetization through $m_y$. For a model with curved Fermi arcs, with a dispersion depending on both $k_y$ and $k_z$, other terms would enter the action in Eq. (56), leading to a different effective action for the magnetization, and in turn a different induced hard-axis anisotropy. The physics remains qualitatively the same.

## A.2 Bulk effective action

We perform the calculation of the bulk effective action in the adiabatic limit, assuming the background magnetization $\tilde{b}_z$ is constant, restoring its $x$-dependence only at the end. We assume a vanishing axial chemical potential $\mu_5 = 0$. The bulk term that appears when $\mu_5 \neq 0$ is discussed in depth in section 3. By integrating out the fermionic degrees of freedom and expanding to second order in the fields $A_\mu$ and $\delta b_\mu$ we get

$$\Gamma^{(2)}[A, b] = \frac{i}{2} \mathrm{Tr}\left( \frac{e\slashed{A} - e\,v_F\,\slashed{\delta b}\gamma^5}{i\slashed{\partial} - e\,v_F\,\tilde{\slashed{b}}\gamma^5} \cdot \frac{e\slashed{A} - e\,v_F\,\slashed{\delta b}\gamma^5}{i\slashed{\partial} - e\,v_F\,\tilde{\slashed{b}}\gamma^5} \right), \tag{62}$$

where the slashed notation is defined as $\slashed{A} = A_\mu \gamma^\mu$, and the trace is over both position space and spin space. We only consider parity odd terms of order $\delta b^2$ and $A\delta b$, which results in one term, namely

$$\Gamma^5_{\mathrm{CS}} = C \frac{\Delta^3}{v_F^3 \hbar^2} \int \mathrm{d}^4 x \; \varepsilon^{\mu\nu\rho\sigma} m_\mu \tilde{m}_\nu \partial_\rho m_\sigma, \tag{63}$$

where $\tilde{m}_\mu = (0, 0, 0, \tanh(x/\lambda))$, and $C$ is a finite, but undetermined coefficient [51, 59–61]. The coefficient is fixed by demanding consistency with the consistent anomaly [28], which in terms of the axial electromagnetic fields takes the form

$$\partial_\mu J_5^\mu = \frac{e^3}{6\pi^2 \hbar^2} \boldsymbol{E}_5 \cdot \boldsymbol{B}_5, \tag{64}$$

where $J_5^\mu$ is the axial vector current for the total system, including contributions of both bound and extended states [28, 62]. The axial current is thus derived by varying both the effective action for the bound states, Eq. (61), and the Chern-Simons term, Eq. (63), with respect to $b_\mu$. By expanding the right hand side of Eq. (64) in terms of the magnetization, the coefficient $C$ is determined, which yields

$$\Gamma^5_{\mathrm{CS}} = \frac{\Delta^3}{12\pi^2 v_F^3 \hbar^2} \int \mathrm{d}^4 x \varepsilon^{\mu\nu\rho\sigma} m_\mu \tilde{m}_\nu \partial_\rho m_\sigma. \tag{65}$$

$\Gamma^5_{\mathrm{CS}}$ does not contribute to the equations of motion of the domain wall as it only renormalizes the Berry phase term in the effective action of the ferromagnet,

$$\Gamma_B = -\frac{2\hbar A}{a^3} \int \mathrm{d}t \; \dot{\phi} X. \tag{66}$$

By expanding $\Gamma^5_{\mathrm{CS}}$ to first order in $X(t)$ around $X(t) = 0$ gives the expression,

$$\Gamma^5_{\mathrm{CS}} = -\frac{A\Delta^3}{9\pi^2 v_F^3 \hbar^2} \int \mathrm{d}t \; \dot{\phi} X, \tag{67}$$

which is indeed of the same form as the Berry phase term in Eq. (66). For system parameter values [3] $|\Gamma^5_{\mathrm{CS}}| \ll |\Gamma_{BF}|$, and the Chern Simons contribution can be omitted.

## A.3 Parity even terms

Besides the parity odd terms, the effective action (or polarization function) will also contain bulk parity even terms, but these will not change the physics in a qualitative way. To illustrate this, these terms can be computed in the adiabatic approximation as was done for the parity odd terms, assuming the background magnetization $\tilde{b}_z$ to be constant, and restoring the $x$

dependence at the end. It was shown in Ref. [63] that the correction due to a constant $\tilde{b}_z$ to the even part of the polarization function is of order $\tilde{b}_z^2$ and breaks gauge invariance. Imposing gauge invariance by using a regularization which preserves it, for example dimensional or Pauli-Villars regularizations, one obtains a zero correction [63,64], so that the even terms of the effective action will not depend on $\tilde{b}_z$. There are in principle two terms, one quadratic in $\delta b_i$ and one coupling $A_\mu$ to $\delta b_i$. The latter, due to the axial nature of $\delta b_i$ which couples with opposite sign to the two chiralities, vanishes, whereas the former is equal to the photon polarization function for a massless Dirac fermion, which is obtained from a standard calculation. Using dimensional regularization and the minimal subtraction renormalization scheme, one has [65]

$$\Gamma_{\text{even}} = \int \frac{d^4 p}{(2\pi)^4} \delta b_i(-p) \Pi^{ij}(p) \delta b_j(p), \tag{68}$$

$$\Pi^{ij}(p) = \frac{\Delta^2}{4\pi^2 v_F^3} \Pi(p^2/\mu_0^2)(p^2 \delta^{ij} + p^i p^j), \tag{69}$$

$$\Pi(p^2/\mu_0^2) = \frac{1}{9\pi}[5 - 3\log(-p^2/\mu_0^2)], \tag{70}$$

with four momentum $p = (\omega, p_i)$. The parameter $\mu_0$ is an energy scale that appears in the process of dimensional regularization. The appearance of this parameter reflects the formal absence of any characteristic scale in the linear electronic spectrum. It could be fixed either experimentally or related to the lattice cutoff of the underlying band structure, by considering a full lattice model of a Weyl semimetal [65].

In the static limit ($\omega \to 0$), $\Gamma_{\text{even}}$ just renormalizes the exchange term of the Heisenberg Hamiltonian. The $\omega$ dependence is, however, quadratic, compared to the linear dependence of the Berry phase term, hence representing a higher order term in a derivative expansion and its effect on the physics can be neglected in a first approximation. Generalizations of the above calculation for finite chemical potential and/or temperature can be obtained from, e.g., Refs. [66,67].

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
