# Peer review of "Electric manipulation of domain walls in magnetic Weyl semimetals via the axial anomaly"

_SciPost Physics, doi:SciPost Phys. 10, 102 (2021)_

## Round 1 · Referee Report · Anonymous · 2021-2-16

Strengths

1 - To my knowledge, this is the first theoretical work to treat the characteristics of domain wall chirality in the context of magnetic Weyl semimetal. This is in a clear contrast with the situation for the domain wall motion, which was discussed to some extent in Refs.17 and 18. In the context of magnetism and spintronics, chiralities of magnetic textures are attracting a great interest, in connection with the emergent electromagnetism and the spin-wave dispersion. The authors' findings may provide new way in manipulating chiralities of magnetic textures more efficiently in magnetic Weyl semimetals. This point meets one of the acceptance criteria, "3. Open a new pathway in an existing or a new research direction, with clear potential for multipronged follow-up work."

2 -The analysis is performed in a systematic way. The authors derive the effective action for the domain wall by the path integral formalism with respect to the electron fields, which is a well-defined and controlled approach. Once the order of perturbative expansion is specified, this approach is capable in deriving all the possible terms in equilibrium. The effective action derived by this approach can be systematically incorporated in the Euler-Lagrange formalism of the domain wall.

3 - The scientific background and the process of analysis are presented in a self-contained manner. In Section 2, the authors introduce the collective coordinate parametrization and its Euler-Lagrange formalism for conventional magnetic domain walls as the scientific background. Such explanation would be helpful for readers who are not familiar with the theory of magnetic textures.

4 - The physical quantities influenced by their findings are properly estimated. The authors suggest that the axial anomaly effect on the domain wall chirality can be achieved with a magnetic field at the order of 0.01-1T and an electric field up to the order of a few MV/m (namely a few volts per micrometer), which are well available in experiments. The estimated domain wall velocity up to 2-3km/s is also in a realistic range that can be measured with the Hall transport or the magneto-optical imaging techniques.

Weaknesses

1 - The derivation of the effective action for magnetic textures and electromagnetic field, using the path-integral formalism, is systematic and straightforward. However, it is capable in capturing only the equilibrium effect, and cannot fully derive the effect from out-of-equilibrium perturbations. For example, the spin-transfer torque (namely the torque from the transport current) would also be present, as pointed out in Ref. 17 and mentioned in Eq.(4), this torque is not captured in the effective action derived in Section 3. To show that the path-integral approach is good enough for their calculation, the authors should clarify their motivation in employing this approach.

2 - I suspect that there are some other equilibrium effects that are missing in the effective Lagrangian. For example, while the authors derive the axial separation effect contribution in Eq.(28), the contribution from the chiral separation effect, namely the axial current $\boldsymbol{J}_5 \propto \mu \boldsymbol{B}$, is missing. Since they are taking into account the external magnetic field, the spin torque induced by this axial current would also be present. The authors should clarify what kind of condition and approximation they have used to achieve Eq.(25).

3 - In Section 4A, there is no intuitive picture why the anomaly effect favors the Bloch-type chirality rather than the Neel-type. Since most of the readers interested in magnetic domain walls may not be familiar with the physics of Weyl fermions, an intuitive picture in parallel with conventional Heisenberg spin systems would be helpful for those readers to understand this important result. The Dzyaloshinskii-Moriya interaction (DMI), namely the noncollinear spin interaction due to the breaking of inversion symmetry, is often responsible for chiralities of magnetic textures. Since $\mu_5$ in this model breaks inversion symmetry, perhaps the chirality effect may be understood as the effective DMI.

4 - The setups employed in Section 4 seem ambiguous. While the magnetic field in Section 4A is fixed in x- and y-directions, it is not specified in Section 4B. Since Eq.(42) is derived from Eq.(31), perhaps B-field is pointing in z-direction, which is the situation quite different from Section 4A. The authors should specify the directions of $\boldsymbol{E}$ and $\boldsymbol{B}$, and $\boldsymbol{B}_5$ corresponding to the domain wall structure, in each section.

5 - In Appendix A1, they identify the bound states at the domain wall, and derive the effective action corresponding to them. While I find this method reliable, I am not confident of the choice of bound states used for the path integral. The bound states shown in Eqs.(A6)-(A7) are the "Fermi arc" states with zero energy, while there are usually many other bound states with finite energies, as shown in Ref.55. The authors should comment why these finite-energy bound states do not contribute to the effective action. For instance, if the domain wall is thin enough, the finite-energy states are energetically well separated from the zero-energy Fermi arc states, and hence the treatment with only the Fermi arc states would be rationalized.

Report

In the present manuscript, the authors focus on the dynamics of domain walls in magnetic Weyl semimetals, and theoretically show how the axial anomaly of the Weyl fermions influences the domain wall dynamics. Starting from the low-energy effective model of Weyl electrons coupled with a domain wall, they derive an effective Lagrangian for the domain wall by integrating out the electron degrees of freedom, and obtain the equations of motion in the collective coordinate formalism. With the obtained equations of motion, the authors mainly find two effects arising from the axial anomaly, which are present if the electric and magnetic fields are applied in parallel: (i) The anomaly leads to the shift of the chirality in the domain wall structure in equilibrium. (ii) In the motion of the domain wall driven by the magnetic field, the anomaly tends to suppress the Walker breakdown, namely the saturation of the domain wall velocity due to the dynamics in the domain wall chirality, and enhances the maximum velocity of the domain wall. The authors expect that these new features would be useful in application of magnetic domain walls to logic gate designs, and also in detecting the axial anomaly directly in experiments.

Throughout this manuscript, I have no doubt that the authors employ the scientifically valid procedure, present their results in a clear way, and cite the previous literatures appropriately. I also find several advantages of this manuscript, as listed in the "Strengths" section. From these points, I consider that this manuscript almost meets the Acceptance criteria of SciPost Physics. On the other hand, I would raise several comments and questions, regarding the validity of their theoretical setups and the significance of their findings, as I list up in the "Weaknesses" section. I would encourage the authors to improve these points before publication.

Requested changes

1 - In Section 4A (left column of page 7), there are statements
"For electric fields larger than this critical value, $E_y > E_c$ ..."
"If the electric field instead is smaller than the critical value ..."
However, I am afraid that the words "large" and "small" are sometimes confusing in this context, since $E_y$ also takes a negative value with large magnitude in their calculations. I would encourage the authors to use some other words.

  • validity: high
  • significance: ok
  • originality: top
  • clarity: good
  • formatting: excellent
  • grammar: perfect

Author:  Julia Hannukainen  on 2021-04-07  [id 1345]

(in reply to Report 1 on 2021-02-16)

We thank the referee for the careful reading of the manuscript and their report. We are glad that they "consider that this manuscript almost meets the Acceptance criteria of SciPost Physics".

Let us address each of the points made by the referee:

1- "The derivation of the effective action for magnetic textures and electromagnetic field, using the path-integral formalism, is systematic and straightforward. However, it is capable in capturing only the equilibrium effect, and cannot fully derive the effect from out-of-equilibrium perturbations. For example, the spin-transfer torque (namely the torque from the transport current) would also be present, as pointed out in Ref. 17 and mentioned in Eq.(4), this torque is not captured in the effective action derived in Section 3. To show that the path-integral approach is good enough for their calculation, the authors should clarify their motivation in employing this approach."

We thank the referee for raising this point. We also respectfully disagree with the referee in that the effective action only captures equilibrium effects. As an example take the simple electric conductivity, $\langle J_i\rangle=\sigma_{ij} E^j$. In the effective action, this manifests as the term $\Gamma=\int d^4x A_i\langle J^i\rangle=\int d^4x A_i\sigma_{ij} E^j$. The electric field is driving the system out of equilibrium, and $\langle J_i\rangle$, captured in the effective action, gives the total local current at a given point in space, consisting in a transport contribution (which is only non-vanishing out-of-equilibtium) and magnetization currents (which can appear in equilibrium when time-reversal symmetry is broken). In general, the effective action contains all the information that can be captured in the Kubo-formulas.

In the present manuscript, we focus on a specific term, Eq. 27, which is only present out-of-equilibrium due to the chiral anomaly, and show how it induces a torque on the magnetization. We of course do not obtain all terms that could appear in the effective action, which include the spin transfer torque term from Ref. 17 (Eq. 4). We comment on what approximations we are taking in the manuscript. To capture the term in Eq. 4 (or Ref. 17), it is necessary to consider explicitly a finite chemical potential, which we do not as it is not necessary to capture the physics of the anomaly. Furthermore, Ref. 17 takes into account scattering from impurities, which we don't. We only include the chemical potential in the relation between $\mu_5$ and $\rho_5$, and scattering from impurities by giving a finite lifetime to axial particles in the anomaly Eq. 18. Again, our focus is to capture the physics of the anomaly, and how it affects the magnetization configuration and dynamics, and our approach and approximations are according to that goal.

We agree with the referee that it was not totally clear, or at least not explicitly stated, that we where taking $\mu=0$ in the computation of the effective action. We address this in the new version.

2- "I suspect that there are some other equilibrium effects that are missing in the effective Lagrangian. For example, while the authors derive the axial separation effect contribution in Eq.(28), the contribution from the chiral separation effect, namely the axial current $J_5\propto\mu B$, is missing. Since they are taking into account the external magnetic field, the spin torque induced by this axial current would also be present. The authors should clarify what kind of condition and approximation they have used to achieve Eq.(25)."

We did not consider the term $J_5\propto\mu B$ because we did not explicitly calculate it with finite $\mu$. The reasons for this are as stated in the previous point. It is not difficult to see that the effect of this term is to renormalize the Zeeman coupling of the magnetic field to the magnetization, Eq. 9. We agree that this should be clarified in the manuscript. We do so in the new version, right after Eq. (30).

3- "In Section 4A, there is no intuitive picture why the anomaly effect favors the Bloch-type chirality rather than the Neel-type. Since most of the readers interested in magnetic domain walls may not be familiar with the physics of Weyl fermions, an intuitive picture in parallel with conventional Heisenberg spin systems would be helpful for those readers to understand this important result. The Dzyaloshinskii-Moriya interaction (DMI), namely the noncollinear spin interaction due to the breaking of inversion symmetry, is often responsible for chiralities of magnetic textures. Since $\mu_5$ in this model breaks inversion symmetry, perhaps the chirality effect may be understood as the effective DMI."

We really thank the referee for pointing this out. As they says, $L_{\mu_5}$ can be seen as an effective DMI, $L_{\mu_5}\propto\textbf{m} \cdot\nabla\times\textbf{m}$. Such terms are known to influence the internal structure or chirality of domain walls. Hence this provides an intuitive picture of why the anomaly, through $L_{\mu_5}$, influences the chirality of the domain wall. We have included a few sentences explaining this at the beginning of section IV (first paragraph). We hope this makes a parallelism with conventional knowledge of Heisenberg spin systems, and provides an intuitive view of the effect.

4- "The setups employed in Section 4 seem ambiguous. While the magnetic field in Section 4A is fixed in x- and y-directions, it is not specified in Section 4B. Since Eq.(42) is derived from Eq.(31), perhaps B-field is pointing in z-direction, which is the situation quite different from Section 4A. The authors should specify the directions of E and B, and B5 corresponding to the domain wall structure, in each section."

We thank the referee for pointing this out; indeed the B-field in section 4B points in the $z$-direction, which we failed to explicitly state. We have now updated the manuscript to include and specifically state the direction of the fields used, and the reasons why, in section 4B.

5- "In Appendix A1, they identify the bound states at the domain wall, and derive the effective action corresponding to them. While I find this method reliable, I am not confident of the choice of bound states used for the path integral. The bound states shown in Eqs.(A6)-(A7) are the "Fermi arc" states with zero energy, while there are usually many other bound states with finite energies, as shown in Ref.55. The authors should comment why these finite-energy bound states do not contribute to the effective action. For instance, if the domain wall is thin enough, the finite-energy states are energetically well separated from the zero-energy Fermi arc states, and hence the treatment with only the Fermi arc states would be rationalized."

We agree with the referee that there are in general also massive (or gapped) bound states, in addition to the gapless Fermi arc. The appearance of gapped bound states depends on the domain wall width, whereas the Fermi arc is always present. The gapped bound states, however, are not 1D, as they disperse in the other dimensions. In this sense, the physics of these bound states is similar to those of bulk states, and one can expect them to contribute just renormalizing the terms of the effective action given by the bulk states. In other words, they are not going to give any new qualitative physics. This is discussed in previous papers, see for example Phys. Rev. B 89, 024413.

The above makes sense in an intuitive way. One can approximate the domain wall to be a Weyl node separation vector that changes sign abruptly, just as a sign function. The physics should not change in a qualitative way just by smoothing this transition between the regions of negative and positive Weyl node separation vectors. It's similar to modeling a boundary. But still one can not do an adiabatic argument, and assume that the Weyl node separation vector is homogeneous, compute things like that, and only insert the space dependence at the end. That way, one can not capture the Fermi arc state, which is in nature non-adiabatic and gives new qualitative physics. So to capture all the physics, one needs to combine the two, adiabatic treatment plus Fermi arc state.

Note that, even if the massive bound states are considered, this would still be an approximation, because the bulk states are being computed as if the Weyl node separation vector was homogeneous (adiabatic treatment). So this could go on and on. The point here is that the exact calculation with the exact states considering the domain wall as a background field is terrifying to say the least (see for example Phys. Rev. D 49, 1980), and not necessary at all to capture the physics.

We mention in the appendix that we use the adiabatic approximation for computing the extended states. We have added a few sentences, right after Eq. A9, saying that there might be also massive bound states, but that these will not give rise to any qualitative new physics.

6- "In Section 4A (left column of page 7), there are statements "For electric fields larger than this critical value, $E_y>E_c$ ..." "If the electric field instead is smaller than the critical value ..." However, I am afraid that the words "large" and "small" are sometimes confusing in this context, since $Ey$ also takes a negative value with large magnitude in their calculations. I would encourage the authors to use some other words."

We thank the referee for pointing this out and have changed this in the updated manuscript.

---

## Round 1 · Referee Report · Anonymous · 2021-3-3

Strengths

1 - A new effect of production of spin torque by the combination of axial anomaly and axial separation effect in a magnetic Weyl semimetal is presented and studied. It is shown that it can be used for magnetic domain wall manipulation by means of the electric and magnetic fields.

2 -Authors study the effects of both bulk and surface states. They show that the latter leads to appearance of hard-wall anisotropy, which causes the delay of Walker breakdown. This increases the maximal velocity of the domain wall.

3 - The paper presents observable signatures of the effect (electric manipulation of the domain wall and controlling of the chirality of the domain wall), which can be studied experimentally.

4 - Presented research has practical application in spintronics. It also shows a new way of studying axial anomaly and related effects in Weyl semimetals experimentally.

5 -The effect presented in the paper can be further investigated, for example in the presence of other configurations of magnetization.

Weaknesses

1 - In Sec. III authors present the effective Lagrangian for the magnetization resulting from the fermionic excitations of the system. As explained in the main text, the main bulk contribution $L_{\mu_5}$ - Eq. (27) - the authors obtain by integrating out the fermionic degrees of freedom up to the second order in magnetization fluctuations. The details of the derivation are said to be relegated to the appendices, yet in appendices A2 and A3, which discuss the bulk effective action, I couldn't find any such derivation. In appendix A2 the only term that arises is a correction to the Berry phase term, which is then omitted, as it has a negligible magnitude. In appendix A3 parity-even terms are considered but, as authors point out, they do not change the physics in a qualitative way. The derivation of term $L_{\mu_5}$, which is the cause of the effect discussed in the paper, appears absent. I think this is a massive omission. From the calculations in the appendices, which are performed, as expected, to the second order in fluctuations, one concludes that term $L_{\mu_5}$ does not exist at all. Its derivation should be presented, and the correct assumptions required to obtain it should be given.

2 - In Eq. (3) the authors introduce an expression for the spin torque. Except for parameters of the model and the chiral current $J_5$, which is given in Eq. (28), it involves a parameter $\rho_s$, the number of magnetic elements per unit volume, the value of which is needed to evaluate the expression. Given that authors obtain the equation of motion of the domain wall, which requires knowledge of the value of the spin torque, I understand the the expression for $\rho_s$ in terms of other parameters of the model is known, and I think it should also be provided to the reader.

3 - Effect discussed in Ref. 17 does not arise in the calculations provided in the paper, however there is no explanation as to why this is the case. I think it should be explained what approximations are being made that make that effect absent or negligible.

4 - In Sec. II authors introduce a domain wall for which the domains' magnetization orientation (z) lies in plane of the domain wall (yz). Is there a reason to assume this very specific geometric relation between the magnetization orientation and domain wall plane? I think a physical reasoning for assuming such a relation should be explained.

5 - The maximal domain wall velocity is given as $\dot X_\text{max}=2.6$ km/s, and it is being pointed out that it is a high value. I think in order to substantiate this claim an estimate of the maximal velocity in the absence of the discussed effect should be added for comparison.

Report

In the paper the authors study axial anomaly and axial separation effect in a magnetic Weyl semimetal, in the presence of a space-dependent magnetization. They show that the combination of those two effects produces a spin torque, which is, to my knowledge, the first time that this mechanism is discussed in the literature, although a different mechanism for producing spin torque in a Weyl semimetal was already proposed in Ref. 17, which is also mentioned in the paper.

This new effect is studied in the presence of a magnetic domain wall, and it is shown that it can be used to manipulate the chirality and the velocity of the domain wall using the electric and magnetic fields. Additionally authors demonstrate that the resulting maximal speed of the domain wall, limited by the occurrence of Walker breakdown, is increased compared to its value without the effect. This happens due to the Fermi arcs bound to the domain wall, which gives rise to hard axis anisotropy, even if none was present in the system intrinsically.

I think that the paper satisfies all the general acceptance criteria of the journal and expectation 3: "Open a new pathway in an existing or a new research direction, with clear potential for multipronged follow-up work", as the presented effect can be studied experimentally, and its experimental signatures are clearly outlined. The motivation for such studies lies both in vast potential applications of the effect in spintronics, and fundamental interest in studying axial anomaly and related effects in Weyl semimetals. From the theoretical side, the effect presented in the paper could be further investigated, for example in the presence of other configurations of magnetization.

I do, however have several remarks, which I listed in the Weaknesses section of the review. I think these have to be addressed before the work can be published, but provided that the paper should be accepted in SciPost Physics.

Requested changes

1 - I believe there is a typo in App. A2 above Eq. A18 in "consistency with the consistent anomaly".
2 - Points listed in Weaknesses section

  • validity: top
  • significance: high
  • originality: good
  • clarity: high
  • formatting: excellent
  • grammar: excellent

Author:  Julia Hannukainen  on 2021-04-07  [id 1346]

(in reply to Report 2 on 2021-03-03)

We thank the referee for their careful review of our work. We will now address the points raised by the referee one by one:

1-"In Sec. III authors present the effective Lagrangian for the magnetization resulting from the fermionic excitations of the system. As explained in the main text, the main bulk contribution $L_{\mu_5}$ - Eq. (27) - the authors obtain by integrating out the fermionic degrees of freedom up to the second order in magnetization fluctuations. The details of the derivation are said to be relegated to the appendices, yet in appendices A2 and A3, which discuss the bulk effective action, I couldn't find any such derivation. In appendix A2 the only term that arises is a correction to the Berry phase term, which is then omitted, as it has a negligible magnitude. In appendix A3 parity-even terms are considered but, as authors point out, they do not change the physics in a qualitative way. The derivation of term $L_{\mu_5}$, which is the cause of the effect discussed in the paper, appears absent. I think this is a massive omission. From the calculations in the appendices, which are performed, as expected, to the second order in fluctuations, one concludes that term $L_{\mu_5}$ does not exist at all. Its derivation should be presented, and the correct assumptions required to obtain it should be given."

We obtain the term $L_{\mu_5}$ from the current given by the axial separation effect, which coefficient we fix by comparing its Feynman diagram to the diagram for the chiral magnetic effect. In the new version we have restructured the argumentation leading to $L_{\mu_5}$, we hope it is more clear now.

In the appendix we calculated at $\mu_5=0$, hence omitting the term $L_{\mu_5}$, because it was already obtained in the main text. In the new version, we mention in the appendix that this term has been obtained in the main text, just for clarity.

2-"In Eq. (3) the authors introduce an expression for the spin torque. Except for parameters of the model and the chiral current $J_5$, which is given in Eq. (28), it involves a parameter $\rho_s$, the number of magnetic elements per unit volume, the value of which is needed to evaluate the expression. Given that authors obtain the equation of motion of the domain wall, which requires knowledge of the value of the spin torque, I understand the the expression for $\rho_s$ in terms of other parameters of the model is known, and I think it should also be provided to the reader."

We thank the referee for raising this point. Indeed the link of the term $L_{\mu_5}$ with the spin torque was not explicitly stated in the previous version. We have added a generic spin torque term $T_e$ to the equations of motion of the domain wall (Eq. 12), and then have given the value of $T_e$ that arises from the contribution $L_{\mu_5}$. By comparison with Eq. 3 now one could give a value for $\rho_s$, but we feel that the important piece missing was giving a explicit value for the induced torque by the anomaly, which is now fixed.

3-"Effect discussed in Ref. 17 does not arise in the calculations provided in the paper, however there is no explanation as to why this is the case. I think it should be explained what approximations are being made that make that effect absent or negligible."

We thank the referee for pointing this out. We don't capture that effect because we don't explicitly introduce a chemical potential in the calculations. This is needed for that term, as it is a Fermi surface term. We don't explicitly compute with $\mu\neq0$ because it is not necessary for the physics that we wanted to capture, this is the physics of the axial anomaly. We have added a paragraph to clarify this point at the end of section III.

4-"In Sec. II authors introduce a domain wall for which the domains' magnetization orientation (z) lies in plane of the domain wall (yz). Is there a reason to assume this very specific geometric relation between the magnetization orientation and domain wall plane? I think a physical reasoning for assuming such a relation should be explained"

The physics of domain walls with magnetic orientations in and out of the domain wall plane can differ considerably due to the coupling to the Weyl quasiparticles. For example, a domain wall with out-of-plane orientation will not posses Fermi arc states localized at the domain wall. It would not generate a net $B_5$ field, and hence the physics of the term $L_{\mu_5}$ would not be present, or at least not in the same manner. We are sure there would still be nice physics in that case, even related to the axial anomaly, but that would require further research and a whole new work. This is something we are presently evaluating.

5-"The maximal domain wall velocity is given as $\dot{X}=2.6Km/s$, and it is being pointed out that it is a high value. I think in order to substantiate this claim an estimate of the maximal velocity in the absence of the discussed effect should be added for comparison."

We agree with the referee and have updated the manuscript with a comparison with a maximal velocity obtained by using a range of typical values for the hardaxis anisotropy for permalloy nanowires. The velocity obtained is between one to two orders of magnitude smaller compared to the velocity we present, which is due to the Fermi arc induced hard axis anisotropy.

6-"I believe there is a typo in App. A2 above Eq. A18 in "consistency with the consistent anomaly"."

This is not a typo; here the expression is to be consistent with the "consistent anomaly". We have added a reference for more clarity.

---

## Round 1 · Referee Report · Anonymous · 2021-3-8

Strengths

(1) The authors show how the joint action of two effects, the
axial anomaly and the axial separation effect, induces
an electric-field-modulated spin torque in domain walls in magnetic Weyl semimetals. This is an important proposal since electric field control of magnetic textures is long sought after in the field of spintronics.

(2) The paper discusses concrete signatures that can potentially be verified in experiments.

(3) Electric field control of magnetic domain walls can also serve as an additional signature of chiral anomaly

Weaknesses

(1) The derivation of Eq. 27 is unclear. It should be expanded in the revised version of the manuscript.

Report

I recommend the paper for publication after it is suitably revised. I believe the journal's acceptance criteria are met. The paper proposes a novel scheme for electric control and manipulation of magnetic textures in Weyl semimetals. This is long sought in the field of spintronics and the authors' proposal to use topological effects such as chiral anomaly to achieve this goal is interesting. The paper has proposals that may trigger concrete experimental efforts. Manipulation of domain walls by electric field in magnetic Weyl semimetals may also provide an additional confirmation of the interesting effect known as chiral anomaly.

  • validity: high
  • significance: high
  • originality: high
  • clarity: good
  • formatting: excellent
  • grammar: excellent

Author:  Julia Hannukainen  on 2021-04-07  [id 1347]

(in reply to Report 3 on 2021-03-08)

We thank the referee for their report. We address the point raised by the referee:

1-"The derivation of Eq. 27 is unclear. It should be expanded in the revised version of the manuscript."

In the new version we have revised the structure of the argumentation leading to $L_{\mu_5}$. We hope it is more clear now.

---

## Round 2 · Referee Report · Anonymous · 2021-4-14

Weaknesses
1 - The derivation procedure of the effective Lagrangian in Section 3 may still appear confusing to some readers. The authors insist that the effective Lagrangian is derived by integrating out the fermionic degrees of freedom from the microscopic Weyl Hamiltonian [Eqs.(23) and (24)]. The terms $L_{\mathrm{bound}}$ and $L_{\mathrm{CS}}^5$ are derived in this way in Appendix A. However, as far as I can see from this manuscript, the term $L_{\mu_5}$ is not derived from path integral, but from the form of the axial separation current phenomenologically. While I have no doubt about this derivation process, I encourage the authors to mention that its derivation process is somewhat different from the other parts of the Lagrangian.
Report
In the revised manuscript, the authors have considered all the suggestions and criticisms raised by the referees, and have significantly improved their presentation. I find that those changes overcome most of the weaknesses pointed out by the referees, except for one raised in this report. I thus believe that this manuscript is worth being published after making some revision.
Requested changes
1 - In the left column of page 7, I find the word "Block" twice, which may be typos for "Bloch".
Author: Julia Hannukainen on 2021-04-28 [id 1387]
(in reply to Report 1 on 2021-04-14)We thank the referee for their report. We address the points raised by the referee:
"1-The derivation procedure of the effective Lagrangian in Section 3 may still appear confusing to some readers. The authors insist that the effective Lagrangian is derived by integrating out the fermionic degrees of freedom from the microscopic Weyl Hamiltonian [Eqs.(23) and (24)]. The terms $L_{bound}$ and $L_{5CS}$ are derived in this way in Appendix A. However, as far as I can see from this manuscript, the term $L_{\mu_5}$ is not derived from path integral, but from the form of the axial separation current phenomenologically. While I have no doubt about this derivation process, I encourage the authors to mention that its derivation process is somewhat different from the other parts of the Lagrangian."
We will update the manuscript by adding the sentence: "The derivation of the second contribution to the effective Lagrangian, $L_{\mu_5}$, differs from the derivation of $L_{\rm{bound}}$; rather than starting from the Weyl Lagrangian in Eq.(23), the existence and specific form of $L_{\mu_5}$ are instead derived from the axial separation effect.", at the beginning of the section deriving $L_{\mu_5}$, in the main text.
"2- In the left column of page 7, I find the word "Block" twice, which may be typos for "Bloch"."
These are indeed typos and will be corrected in the updated manuscript.

---

## Round 2 · Referee Report · Anonymous · 2021-4-21

Report
In their resubmission authors have addressed well all the point raised in my first report, and I think the manuscript makes for a clear and interesting read, to which I have no further remarks. I stand by my earlier assessment that the paper satisfies all the general acceptance criteria of the journal and expectation 3: "Open a new pathway in an existing or a new research direction, with clear potential for multipronged follow-up work", as I justified in the previous report. Therefore, I recommend the manuscript for publishing in SciPost Physics.

---

## Round 2 · Author Response

List of changes
Section I:
Redefined the constants of three inline equations in the paragraph following Eq. 1.
Section II.
Added a term to Eq. 12, and described it in the text below this equation together with a reference.
Section III.
Rewritten the section describing the origin of $L_{\mu_5}$. This section starts at the next new paragraph following Eq. 26, and ends with Eq. 28.
Added an equation for the spin Torque; Eq. 30.
Added a paragraph directly after Eq. 30 describing a term due to the chiral separation effect, and why it is neglected.
Added a paragraph in the end of the section describing why we do not capture the term in Eq. 4.
Section IV.
Added a comment on the resulting chiralities; first paragraph in the section, sentence starting with: "Intuitively, the effect can be understood..."
Subsection A.
Rewritten the paragraph following Eq. 42, describing the minimum energy configuration due the field $E_y$.
Subsection B.
First paragraph: Added a description of the choice of magnetic field.
Eq. 44: Rewrote the scalar product in terms of the only non-zero term: $E_zB_z$
Last part of the paragraph after Eq. 45: added a section comparing the value of the velocity due to the effect from the Fermi-arcs, to typical domain wall velocities in nanowires.
Appendix.
After equation A2. Added a sentence explaining that the chemical potential is taken to be zero.
Directly following equation A9: Added a comment on additional gapped bound states.
Typos:
Caption in Figure 3: Changed $B_x$ to $B_y$.
Changed sign of $\gamma^5$ in Eq. 24 and Eq. A2.
Changed $\gamma_5$ to $\gamma^5$ in Eq. A.16.

---

## Round 2 · List of Changes

Section I:
Redefined the constants of three inline equations in the paragraph following Eq. 1.
Section II.
Added a term to Eq. 12, and described it in the text below this equation together with a reference.
Section III.
Rewritten the section describing the origin of $L_{\mu_5}$. This section starts at the next new paragraph following Eq. 26, and ends with Eq. 28.
Added an equation for the spin Torque; Eq. 30.
Added a paragraph directly after Eq. 30 describing a term due to the chiral separation effect, and why it is neglected.
Added a paragraph in the end of the section describing why we do not capture the term in Eq. 4.
Section IV.
Added a comment on the resulting chiralities; first paragraph in the section, sentence starting with: "Intuitively, the effect can be understood..."
Subsection A.
Rewritten the paragraph following Eq. 42, describing the minimum energy configuration due the field $E_y$.
Subsection B.
First paragraph: Added a description of the choice of magnetic field.
Eq. 44: Rewrote the scalar product in terms of the only non-zero term: $E_zB_z$
Last part of the paragraph after Eq. 45: added a section comparing the value of the velocity due to the effect from the Fermi-arcs, to typical domain wall velocities in nanowires.
Appendix.
After equation A2. Added a sentence explaining that the chemical potential is taken to be zero.
Directly following equation A9: Added a comment on additional gapped bound states.
Typos:
Caption in Figure 3: Changed $B_x$ to $B_y$.
Changed sign of $\gamma^5$ in Eq. 24 and Eq. A2.
Changed $\gamma_5$ to $\gamma^5$ in Eq. A.16.

---

## Editorial Decision

published